# Identification and Characterisation of a Myxoma Virus Detected in the Italian Hare (*Lepus corsicanus*)

**DOI:** 10.3390/v16030437

**Published:** 2024-03-12

**Authors:** Elisa Rossini, Moira Bazzucchi, Valter Trocchi, Francesca Merzoni, Cristina Bertasio, Sascha Knauf, Antonio Lavazza, Patrizia Cavadini

**Affiliations:** 1Istituto Zooprofilattico Sperimentale della Lombardia e dell’Emilia Romagna “Bruno Ubertini” (IZSLER), Via Antonio Bianchi, 7/9, 25124 Brescia, Italy; elisa.rossini@izsler.it (E.R.); francesca.merzoni@izsler.it (F.M.); cristina.bertasio@izsler.it (C.B.); antonio.lavazza@izsler.it (A.L.); patrizia.cavadini@izsler.it (P.C.); 2WOAH Reference Laboratories for Myxomatosis and for RHD, Via Antonio Bianchi, 7/9, 25124 Brescia, Italy; 3Federazione Italiana della Caccia (FIDC), Via Garigliano 57, 00198 Roma, Italy; valter.trocchi@fidc.it; 4Institute of International Animal Health/One Health, Friedrich Loeffler Institut, Südufer 10, 17493 Greifswald-Insel Riems, Germany; sascha.knauf@fli.de; 5Professorship for One Health/International Animal Health, Faculty of Veterinary Medicine, Justus Liebig University, 35392 Giessen, Germany

**Keywords:** poxvirus, myxomatosis, *L. corsicanus*, diagnosis, NGS

## Abstract

Myxoma virus (MYXV) is a Leporipoxvirus (genus) belonging to the family Poxviridae; it is characterised by a genome of approximately 161 kb dsDNA encoding for several proteins that play an essential role in both host spectrum determination and immunomodulation. The healthy reservoir of the virus is *Sylvilagus* spp. At the same time, in wild and domestic European rabbits (*Oryctolagus cuniculus*), MYXV is the etiologic agent of myxomatosis, a disease with an extremely high mortality rate. In 2014, an interspecies jump of MYXV was reported in *Lepus europaeus* in the UK. In 2018, myxomatosis induced by a new recombinant strain called MYXV-To was identified during a large outbreak in Iberian hares (*Lepus granatensis*) in Spain. Here, we describe the case of myxomatosis in another hare species: an adult male Italian hare (*Lepus corsicanus*) found dead in 2018 in Sicily with lesions suggestive of myxomatosis and treponema infection. Laboratory tests, e.g., end-point PCR and negative staining electron microscopy, confirmed the presence of both pathogens. MYXV was then isolated from tissue samples in permissive cells and sequenced using NGS technology. Main genomic differences concerning known MYXV strains are discussed.

## 1. Introduction

Myxomatosis is an endemic and highly lethal viral disease of European rabbits (*Oryctolagus cuniculus*) caused by infection with Myxoma virus (MYXV; genus Leporipoxvirus—Chordopoxvirinae subfamily) [1]. Its genome contains a large linear double-stranded DNA, consisting of 158 unique open reading frames, 12 of which are diploid in terminal inverted repeats (TIRs) [2]. A characteristic brick-shaped virion contains the genome, and viral replication occurs exclusively in the cytoplasm of infected cells [3].

In the European rabbit, MYXV occurs in at least two clinical forms: (i) the ‘classical’ or ‘typical’ myxomatosis is mainly observed in wild and pet rabbits, which are usually infected by biting arthropods such as mosquitoes and fleas, and (ii) the respiratory form (amyxomatous or ‘atypical’ myxomatosis), that is more frequent in farmed rabbits in which the viral transmission more frequently occurs by direct contact with infected rabbits [4]. Classical myxomatosis is characterised by large skin nodules (myxomas) and is a more aggressive and lethal disease with high mortality rates (<80%) [5]. On the contrary, atypical myxomatosis is generally nonlethal; animals show different, mild skin lesions and prominent respiratory signs frequently caused by secondary pathogens (e.g., *Pasteurella multocida*).

After infection, MYXV can be detected in the lymph nodes, draining the inoculation site, where it replicates within the lymphocytes of the T cell zone. Then, from the lymph nodes, the virus disseminates to other tissues such as the lung, testis, spleen, and mucocutaneous sites (nose, conjunctivae, and skin) [6]. The death following myxoma virus infection is mainly due to secondary bacterial infection and septicaemia favoured by virus-induced immunosuppression. Overall, the clinical signs depend on the virulence of the virus strain, the inoculation route, and the host immune status. In contrast, a single cutaneous lesion, restricted to the point of inoculation, is seen in MYXV natural long-term host, i.e., the South American *Sylvilagus brasiliensis* and the North American *Sylvilagus bachmani* [7].

In Europe, the disease spread after the deliberate release of the MYXV strain (Brazil Campinas/1949), now called the Lausanne (Lu) strain, caused by a landowner in France [8]. This occurred in June 1952, and from this starting point, MYXV spread in the wild and domestic rabbit populations of Europe, remaining phylogenetically related to the initial viral strain [8].

Although myxomatosis is known to primarily affect European rabbits, clinical manifestations associated with MYXV infection have been sporadically reported in other European leporids. While Mountain hares (*Lepus timidus*) are known to be occasionally infected, naturally or experimentally [9], in 2014, a single case of myxomatosis was detected in a European brown hare (*Lepus europaeus*) in the UK [10]. Then, a cross-species jump to the Iberian hare (*Lepus granatensis*) was observed in Spain and Portugal in 2018 [11]. At that time, a new MYXV strain (MYXV-To), resulting from a recombinant event between a MYXV and a still-unreported poxvirus, was isolated and genotyped. Compared to the reference strain Lausanne (MYXV-Lu), MYXV-To encodes three disruptive genes (*M009L, M036L*, and *M152R*) and presents an insertion of ~2.8 kb within the *M009L* gene [12,13]. In this recombinant “cassette”, four new poxvirus genes were found, among which the *M064R*-like gene seems to have a possible function in the interactions of the host with this new recombinant poxvirus. 

The present study aims to describe one case of myxomatosis in a further hare species, i.e., Italian hare (*Lepus corsicanus*), and characterise the isolated MYXV strain in comparison with the already detected MYXV strains in other leporid species.

## 2. Materials and Methods

### 2.1. Sampling and Pathology

In November 2018, an adult male Italian hare (*Lepus corsicanus*) was found dead in Racalmuto, Agrigento province—Sicily (Italy). At necropsy, the lesions were compatible with those typically detected in European rabbits with myxomatosis (Figure 1). Moreover, the oral and genital lesions were also potentially consistent with syphilis caused by *Treponema paraluisleporidarum ecovar Lepus* (in this publication, we use the species name *paraluisleporidarum* synonym to *paraluiscuniculi*), a non-zoonotic sexually transmitted agent [14]. Therefore, for laboratory examination, tissue samples were taken from the peri-ocular, labial, auricular skin, and eyelids and stored at −80 °C.

### 2.2. Viral Isolation

Viral isolation has been set up in vitro using rabbit kidney cell line RK13 (CCL-37, ATCC, Manassas, VA, USA), maintained as indicated by the manufacturer. Cells were seeded in 24-well plates (90,000 cells/well), and twenty-four hours later, plates were infected with 200 μL/well of homogenate from ear/eyelid lesions at serial dilution (10^−1^, 10^−2^, 10^−3^). After an incubation of two hours, cells were washed and grown in a complete medium with 1% FBS. Each infection was tested in duplicate, and the virus cytopathic effect (CPE) was observed using an inverted microscope until five days post-infection. Five days post-infection, the cells were also fixed with acetone solution 80% (*w*/*v*) (Merck Serono, Milan, Italy) for 20 min at −20 °C, washed with PBS 1X, and incubated for 30 min at 37 °C with a hyperimmune anti-myxoma virus serum, FITC conjugated. The hyperimmune serum was produced at IZSLER in 2007 from rabbits vaccinated with Nobivac (Intervet International BV, Boxmeer, The Netherlands) and challenged with a “field” strain isolated from wild rabbit in Cambridgeshire (UK) in 1999. After rinsing in PBS, cells were observed by a fluorescence microscope. To proceed with the virus isolation and for performing electron microscopy (EM) observation, the infected cells not fixed with acetone were frozen–thawed at −80 °C and 37 °C twice to release the viruses from the infected cells, thus harvested and centrifugated. Finally, the viral stock was collected at the fourth passage and stored at −80 °C. The isolated strain was named MYXV-Ag.

### 2.3. Electron Microscopy

EM examination of skin specimens and supernatants of infected RK13 cells showing an evident cytopathic effect was performed using the “drop method” and the direct pelleting of viral particles on grids (“Airfuge method”). The first consists of observing a viral suspension obtained directly by the homogenisation of small pieces of tissues in a mortar with a few drops of phosphate-buffered saline, pH 7.2. Carbon-coated grids were floated on a drop of the viral suspension for 10 min. The second method was performed according to that previously reported [15]. Samples were frozen and thawed twice and centrifuged at 4000× *g* for 20 min and 9300× *g* for 10 min to clarify the supernatant. The second supernatant (82 μL) was ultra-centrifuged in an Airfuge centrifuge (Beckman Coulter, Brea, CA, USA) for 15 min at 21 psi (82,000× *g*), fitted with an A 100 rotor that held six 175 μL test tubes containing specific adapters for 3 mm grids, which enables direct pelleting of virus particles on grids. Carbon-coated, formvar copper grids were stained with 2% sodium phosphotungstate, pH 6.8, for 1.5 min, and observed with a TEM FEI Tecnai G2 Biotwin (Thermo Fisher, FEI, Hillsboro, OR, USA) operating at 85 kV. 

### 2.4. DNA Extraction and PCR Amplification

*Treponema paraluisleporidarum ecovar Lepus* infection diagnosis was conducted using molecular biology techniques described in Hisgen et al. [16]. An end-point PCR was performed to diagnose the myxoma virus infection. The viral DNA for the molecular analysis was extracted from each of the four sampled tissues (peri-ocular, labial, auricular skin, and eyelids) according to Cavadini et al. [17,18] and was indagated for a portion of approximately 470 bp of the *M071R* gene conserved in MYXV. Both positive and negative extraction and amplification controls were included. To confirm the NGS data and to exclude the presence of MYXV-To, we developed a new end-point PCR near the insertion region in *M009L*, which is able to discriminate a MYXV-Lu-like strain from MYXV-To. PCR conditions were the following: an initial activation step at 98 °C for 30 s, 35 cycles at 98 °C for 10 s, 60 °C for 30 s, 72 °C for 1.5 min, followed by a final extension of 7 min at 72 °C. Primer sequences were reported in Table 1.

### 2.5. Whole Genome Sequencing

To sequence the complete genome of the MYXV-Ag, DNA was extracted from 200 μL of viral supernatant at the fourth isolation step. The sequencing library was obtained with the DNA Prep (M) Tagmentation Library Preparation kit (Illumina) and sequenced on the Illumina MiniSeq platform (2 × 150 bp). All bioinformatics analyses were conducted on Galaxy platform [19]. The quality of the reads was assessed using FastQC (Galaxy Version 0.74+galaxy0) [20] and trimmed using Trimmomatic (Galaxy Version 0.39+galaxy0). Reads were assembled de novo using the SPAdes assembler [21] (Galaxy Version 3.15.4+galaxy1) using default parameters. The contigs containing MYXV genomic DNA were identified using Bowtie 2 (Galaxy Version 2.5.0+galaxy0) and ordered into a single genome sequence against the Myxoma virus isolate Lausanne genome [8] (KY548791, MYXV-Lu, 161,773 bp) using Minimap2 (Galaxy Version 2.26+galaxy0) and mapper (Galaxy Version 0.5.2). Only one complete copy of the TIR was assembled at either the 5′ or the 3′ end. Contigs mapping at the opposite end and overlapping up to a full read length of the complementary TIR were observed following identification of the TIR junction. We duplicated the complete TIR and generated a reverse complement of the added sequence on the opposite end. However, to ensure that the TIR regions were correctly recreated and the junctions were accurate, we sequenced them individually by performing two LONG PCR (primers shown in Table 1), anchoring the amplicon on the unique sequences of a core genomic region. LONG PCR’s expected amplicons were 12,300 bp (left TIR) and 12,600 bp (right TIR).

Amplification of the two TIRs was performed using Invitrogen Platinum SuperFi PCR Master Mix (Invitrogen, Waltham, MA, USA), following the manufacturer’s instructions. The PCR products of the LONG PCRs were purified with NucleoSpin Gel and PCR Clean-up kit (Mecherey-Nagel, Dueren, Germany), and the purified DNA was sequenced using the NGS technology described above. Trimmed TIRs reads were assembled de novo with SPAdes assembler (Galaxy Version 3.15.4+galaxy1), MIRA [22], and Velvet [23] in Shovill (Galaxy Version 1.1.0+galaxy2). The obtained contigs were aligned with the hypothetical genome, and the final assembly was verified by remapping the reads to the assembly using Bowtie2 (Galaxy Version 2.5.0+galaxy0). The resulting BAM file was visualised by UGENE ver.44.0.

Genome annotation was transferred from the MYXV strain Lausanne AF170726, firstly annotated and sequenced by Cameron and colleagues in 1999 [2], to the newly sequenced MYXV genomes using GATU (https://4virology.net/virology-ca-tools/gatu/) (accessed on 7 March 2024).

The assembled sequence of MYXV-Ag was deposited in GenBank with accession number PP105561.

## 3. Results

### 3.1. Viral Isolation

The successful isolation of MYXV-Ag in RK13 cells from separate ear/eyelid samples confirmed its viability and infectiousness, proving that the virus was multiplying and had spread to various hare tissues. Viral isolation was confirmed by the presence of CPE at day five in RK13 sub-confluent infected cells and by immunofluorescence of the cells. The characteristic CPE at an early stage of infection was observed from day three after inoculation and reached its maximum five days post-infection (Figure 2). Unlike untreated cells (Figure 2a), virus-infected cells showed an elongated morphology more evident at high virus concentrations (Figure 2b). To demonstrate the presence of the viral particles in the RK13 cells, in addition to EM, an immunofluorescence analysis was performed on day five (Figure 2c,d), which confirmed the presence of Myxoma virus foci [24]. Lowering the concentrations of the tissue extract used as inoculum resulted in a reduction or disappearance of these morphological alterations.

### 3.2. Electron Microscopy

Based on their typical morphologic characteristics, we identified the viral particles at magnifications of 11,000×–26,500×. The negative staining EM examination of both the skin samples and the cell culture supernatants revealed the presence of viral particles morphologically related to the genus orthopoxvirus (Figure 3). The particles exhibited the typical bricklike shape, and their size was approximately 320 × 235 nm (range 300–350 × 220–250). The virions observed in the skin samples presented, in an almost equal number, both the known morphological aspects: the “M” forms showing the typical ribbon structure of the viral surface and the “C” forms, slightly bigger, with a uniform electron density and a thick capsule outlined by a ragged edge [25]. In cell culture supernatants, only “fresh” M particles were observed.

### 3.3. Molecular Characterisation 

The positivity of Treponema paraluisleporidarum in samples taken from oral and anogenital lesions has already been described by Hisgen et al. [16]. At the same time as this diagnosis, we performed a PCR amplification specific for the myxoma virus genome from each sampled tissue. All tested samples were positive, showing the 470 bp band for the *M071R* gene. Moreover, to exclude the presence of the MYXV-To strain, the pool of the four tissue samples was tested for the characteristic insertion of the *M009L* gene. The resulting amplicon of 316 bp confirmed the presence of just a MYXV-Lu-like field strain, excluding a MYXV-To-like strain (Figure 4). 

### 3.4. Whole Genome Sequencing 

The complete genome of MYXV-Ag, reconstructed as explained above, consists of 161,634 bp (mean coverage 150×), 170 genes, of which 146 are unique and 12 duplicates in TIR. Table 2 shows the differences between MYXV-Lu (KY548791), the genome used to order contigs, and MYXV-Ag (this study).

Some differences were found in genes related to host range determination or virulence (*M002L/R, M005L/R, M064R, M148R, M150R, M153R, M156R*). As already described for other viral strains, MYXV-Ag also shows changes in the length of the intergenic region between *M002L/R* and *M003.1L/R* [26], while many single nucleotide variants (SNVs) occur in the coding sequences. Some SNVs are nonsynonymous, giving rise to non-conservative amino acid substitution. Unfortunately, we cannot know whether these impact the biological function of the proteins themselves. There are no frameshift mutations. Many SNVs are unique for this virus, and few others have already been described. For example, the deduced 204 aa sequence of host range factor M64 differs from MYXV-Lu for the insertion of a glutamic acid (▼E164) between the aa 163–164 that have been already described in some grade 1 Australian isolate like Wellington (JX565582) [27], but it also shows a unique, non-conservative D173G. Instead, the P173S replacement in M150 had already been found in Australian viruses spanning the virulence spectrum [27].

We found nine nt deletions in *M153R*, but it does not change the reading frame. The M153 protein is missing three amino acids from that described in MYXV-Lu (Δ176–178). Instead, the M156 protein loses the entire first N-term part consisting of 27 aa due to one nucleotide transition (T→C, nt 149,927) that disrupts the first ATG in the *M156R* gene, establishing a similar situation to that already described in MYXV-MSW (accession number KF148065) and the orthologous genes of other poxviruses [28].

## 4. Discussion

Poxviruses encode a group of genes to maintain or expand their host tropism through a still elusive mechanism [6]. In particular, the MYXV genome encodes specific proteins that act, in susceptible hosts, as immune modulators of the immune system and/or inhibitors of apoptosis while also interfering with chemotaxis and leukocyte activation [6]. It could be speculated that, in the case described here in an adult Italian hare individual, the MYXV found a host, already defiled by *Treponema paraluisleporidarum* infection, more susceptible to becoming infected and developing clinical disease. Furthermore, the simultaneous presence of the two pathogens likely created a situation favourable to increased replication of both pathogens, causing a synergic effect in terms of disease development. Impairment of the immune system could thus have facilitated and allowed, in a host not susceptible under normal conditions, the dissemination of MYXV and the subsequent appearance of signs of the disease.

Our hypothesis fits concepts reported by Haller and colleagues [29], who affirm that differences in host range genes between poxviruses can influence the host range. Such differences may lead to different coding capacities that could directly affect interactions with host molecules or alter protein stability. In addition, genetic variations among different viral strains may also lead to altered levels of transcription or translation. The authors also argue that host-specific differences influence the expression of host viral factors; this could fit well with the situation described in the MYXV + Treponema coinfected Italian hare. On the host side, the presence or absence of antiviral genes or their expression levels could directly influence viral replication, and some poxviruses that exhibit species- or cell-type specificity may require certain host molecules to replicate successfully. Species-specific variations in host antiviral genes and differences in their expression could directly influence interaction with host proteins [29].

Therefore, to better understand the epidemiological finding, we sequenced the whole viral genome by NGS to investigate whether any significant differences in host spectrum genes were present. As pointed out in the results section, an essential gene in determining the host range, the *M156R*, appeared to have significant differences from the strains generally detected in Italy or Europe [8].

In MYXV strains originating from South America, such as MYXV-Lu, which originally comes from Brazil, the *M156R* is annotated to encode a 102-amino acid protein with a predicted molecular mass of 12 kDa [2,30]. The M156 orthologs of the closest relatives, rabbit fibroma virus and the Californian MYXV MSW strain, lack this putative extension but contain a putative start codon that encodes for 78- and 77-aa-long ORFs, respectively, with predicted molecular masses of 9 kDa [28]. A putative start codon at the corresponding position is also found in the South American-derived MYXV strains. The nucleotide transition occurring in the first putative *M156R* ATG could allow coding a shorter protein isoform, much more similar to that described in the other poxvirus Vaccinia Virus (VACV) K3 orthologs. We could not exclude the short isoform (75 aa) translation and function preservation also because it seems to be under the control of the same early promotor already preserved in the other poxvirus coding for the ortholog [28]. In addition, Peng and colleagues [30] indicate that the short form of M156 seems to be the predominant translated form also in the presence of the first start codon.

The innate immune response provides the first line of host defence; it includes antiviral proteins that can lead to the direct elimination of viruses or induce the expression of type I interferons (IFNs), proinflammatory cytokines, chemokines, and other antiviral proteins [31,32,33]. Poxviruses encode diverse PKR antagonists that either directly or indirectly inhibit the Protein kinase R (PKR) pathway. In VACV, these proteins are encoded by E3L, K3L, D9R, and D10R [33], and the M156 protein is the K3 orthologue as reported before.

Yu and colleagues [34] demonstrated that the maladaptation after a virus–host switch leads to increased activation of the proinflammatory NF-κB pathway and can result in substantially different immune responses in aberrant hosts. These different host responses may contribute to altered viral dissemination and influence viral pathogenesis. Notably, they also conducted their experiments by considering the short form of the M156 produced by MSW MYXV and its natural host, *Sylvilagus bachmani.* We cannot exclude that in the *Lepus corsicanus* described here, the concomitance of dual infection and the presence of an M156 short form have played a synergistic role in MYXV dissemination.

## 5. Conclusions

The presence in Sicily of numerous sympatric populations of Italian hares, an endemic species of southern-central Italy and Sicily, and European wild rabbits, a species native to the Iberian Peninsula introduced in historical times in Sicily, requires suitable health monitoring activities and epidemiological studies, which are important for the conservation of populations. For the Myxomatosis, it will also be interesting to define the prevalence of the infection by performing serological studies both by competitive ELISA and comparative virus serum-neutralisation (VN) by using different strains, i.e., the one here isolated from the Italian hare and others of the MYXV wild type circulating in the same region and the rest of Italy. The combination of these data will help determine whether this is an isolated case or new evidence of host adaptation.

## Figures and Tables

**Figure 1 viruses-16-00437-f001:**
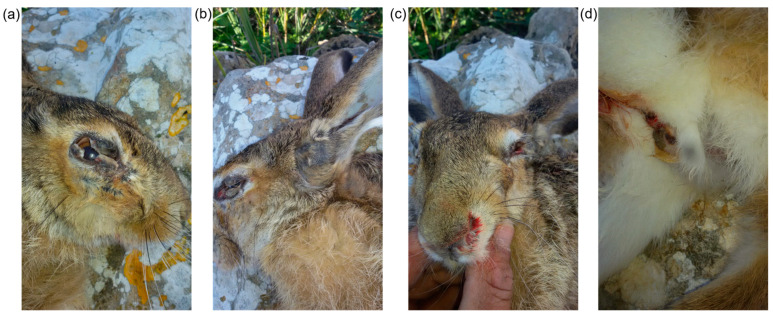
**Adult male Italian hare lesions.** Lesions are compatible with Myxomatosis and *T. paraluisleporidarum ecovar Lepus* infection. (**a**–**c**) Blepharoconjunctivitis; (**d**) genital lesion.

**Figure 2 viruses-16-00437-f002:**
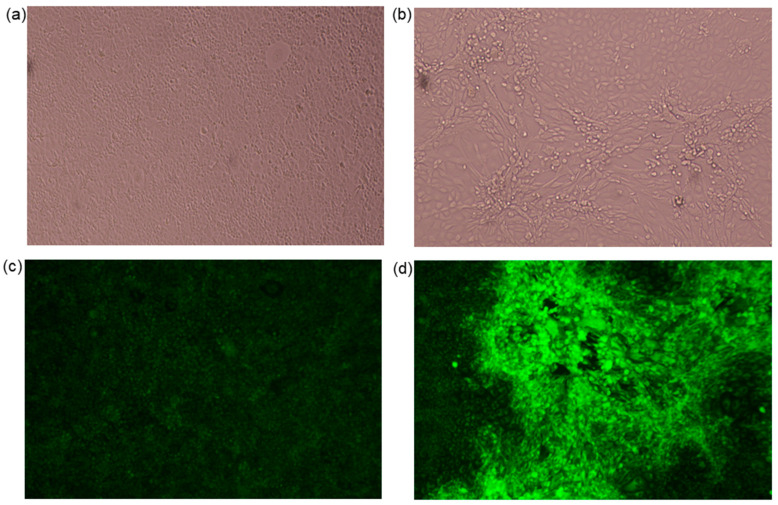
**Images of myxoma virus (MYXV)-infected cells**. RK13 cells were infected with MYXV-Ag. Five days post-infection, the virus-infected cells were visualised with an inverted microscope in the absence (**left image**) or presence of MYXV (**right image**). Live imaging conditions in control (**a**) and infected cells (**b**). Immunofluorescence of control (**c**) and infected cells (**d**). Images were acquired using a direct microscope (Leica Microsystems, Wetzlar, Germany) at 10× magnification.

**Figure 3 viruses-16-00437-f003:**
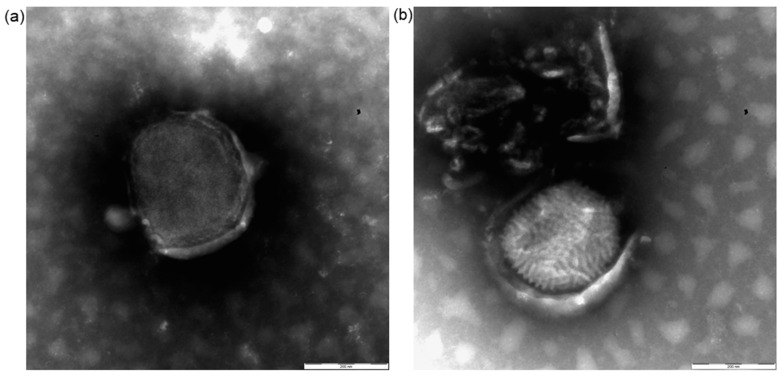
**Images of negative staining electron microscopy (EM).** (**a**) “C” form viral particle; (**b**) “M” form viral particle. Bar = 200 nm.

**Figure 4 viruses-16-00437-f004:**
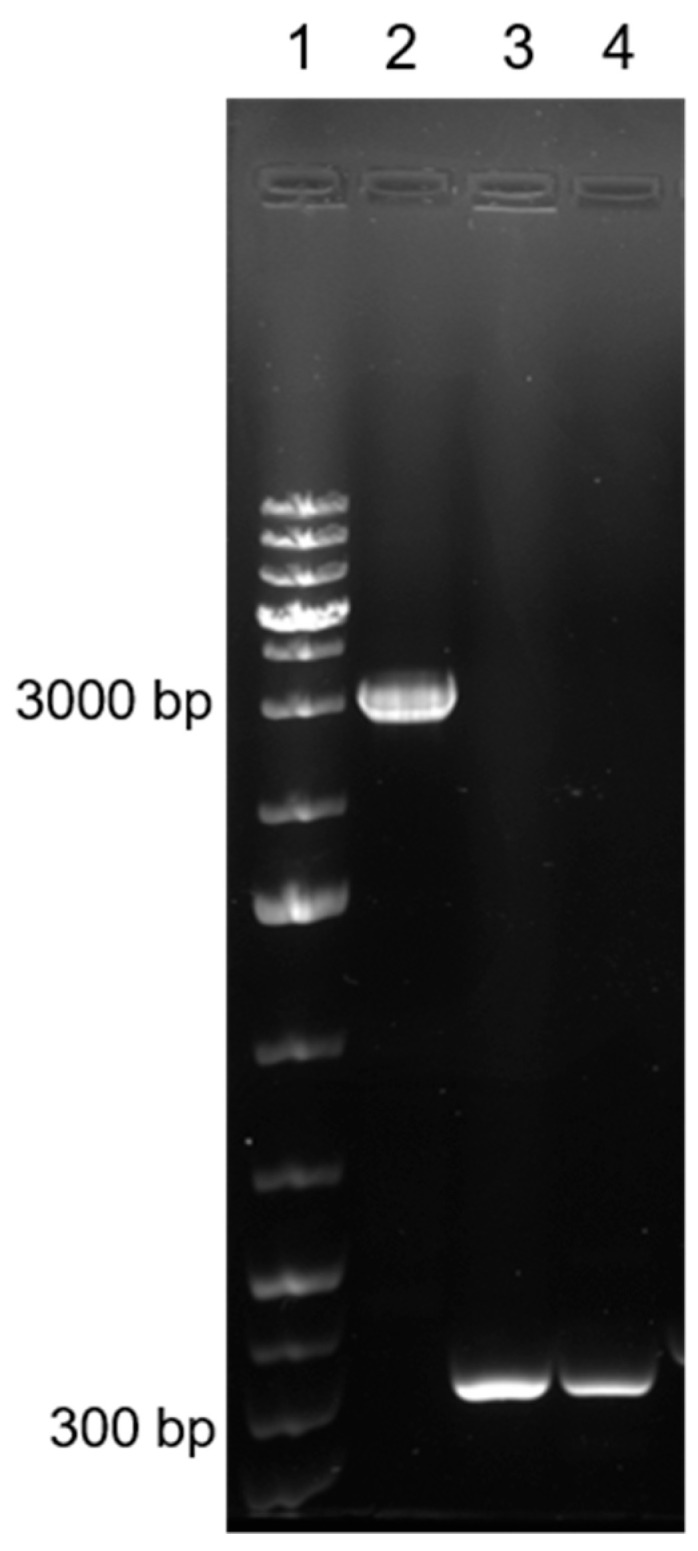
**Discriminatory PCR between MYXV-Lu-like strain and MYXV-To strain.** Ten microliters of PCR product were run on 1% agarose gel and stained with Eurosafe Nucleic acid staining solution. Lane 1: GeneRuler 1 kb Plus DNA Ladder (Invitrogen), lane 2: MYXV-To strain, lane 3: MYXV-Lu-like field strain, lane 4: MYXV-Ag.

**Table 1 viruses-16-00437-t001:** Primer pairs for molecular biology study.

Sequences of Primers for MYXV Infection Diagnosis			
Primer ID	Sequence (5′-3′)	Position nt ^a^	Expected Amplicon (bp)
M071-F	ACCCGCCAAGAACCACAGTAGT	67,229–67,250	272
M071-R	TAACGCGAGGAATATCCTGTACCA	67,700–67,677	
M009-F	ATACACGCCGACGCATTACG	12,181–12,200	316
M009-R	ACGAGAGATACGCTGAAGAAC	12,496–12,476	
**Sequences of primers for MYXV TIR amplification**			
M0005-F	ACGCGGAAGTCTGCCTATTT	230–249	12,266
M009-R	ACGAGAGATACGCTGAAGAAC	12,496–12,476	
M153-F	CATTTATGGTATCCGCATTAAC	148,944–148,965	12,600
M0005-F	ACGCGGAAGTCTGCCTATTT	161,544–161,525	

^a^ Nucleotide position based on the MYXV-AF170726 sequence.

**Table 2 viruses-16-00437-t002:** Myxoma virus (MYXV)-Ag differences from MYXV-LU.

Gene Name	Function of Gene Product	Start	Stop	+/−	Gene Size	Protein Size	MYXV-Lu Protein Size	% aa Similarity	aa Differences
*M002L*	Tumor necrosis factor receptor (TNF-R) homologue	1664	2644	−	981	326	326	99.7	A131V, E224K
*M005L*	E3 Ub ligase	4864	6315	−	1452	483	483	99.8	V320A
*M012L*	dUTP nucleotidylhydrolase	14,070	14,516	−	447	148	148	99.3	K47T
*M031R*	Virosome protein	30,071	31,252	+	1182	393	393	99.7	R75H
*M032R*	VACV E6R	31,262	32,959	+	1698	565	565	99.8	G190V
*M034L*	DNA Pol	33,777	36,797	−	3021	1 006	1 006	100.0	V620I
*M036L*	VACV O1L/Erk1/2 signaling?	37,142	39,184	−	2043	680	680	99.7	R95C
*M044R*	RNA helicase NPH-II	44,090	46,126	+	2037	678	678	99.9	P82S
*M064R*	Host range protein	59,564	60,178	+	615	204	203	99.0	▼164E, D173G
*M079R*	Uracil-DNA glycosylase	76,264	76,920	+	657	218	218	99.5	A183T
*M092L*	Core protein	89,899	91,860	−	1962	653	653	99.8	S85L
*M094R*	RNA Pol subunit	92,414	92,908	+	495	164	164	99.4	D15G
*M096L*	Early transcription factor subunit	94,054	96,189	−	2136	711	711	100.0	N472S
*M099L*	Core protein precursor	97,331	100,036	−	2706	901	901	99.9	R249H
*M109L*	VACV A19L	105,676	105,897	−	222	73	73	98.6	G6E
*M116L*	IMV membrane protein	113,227	113,649	−	423	140	140	99.3	A139T
*M127L*	Photolyase	119,653	120,990	−	1338	445	445	99.6	R207H, A257P
*M138L*	Alpha-2,3 sialyltransferase	133,811	134,683	−	873	290	290	99.7	A15T
*M147R*	Ser/Thr protein kinase	140,637	141,503	+	867	288	288	100.0	H162Y
*M148R*	Putative E3 Ub ligase	141,564	143,591	+	2 028	675	675	99.9	P643S
*M150R*	E3 Ub ligase; NF-κB inhibition	145,129	146,613	+	1485	494	494	99.8	P173S
*M152R*	SERP-3	147,626	148,426	+	801	266	266	99.6	R218C
*M153R*	E3 Ub ligase/MHC-1 downregulation	148,464	149,075	+	612	203	206	98.5	Δ176–178
*M156R*	Interferon resistance; elF2α homologue	150,007	150,234	+	228	75	102	73.5	short form
*M005R*	E3 Ub ligase	155,320	156,771	+	1 452	483	483	99.8	V320A
*M002R*	Tumor necrosis factor receptor (TNF-R) homologue	158,991	159,971	+	981	326	326	99.7	A131V, E224K

## Data Availability

The genome sequence of MYXV-Ag was deposited in the NCBI GenBank database under accession number PP105561.

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
