# Peer review of "Identification and Characterisation of a Myxoma Virus Detected in the Italian Hare (Lepus corsicanus)"

_viruses, 2024, doi:10.3390/v16030437_

Round 1

Reviewer 1 Report

Comments and Suggestions for Authors

Manuscript ID: viruses-2892949

“Identification and characterisation of a Myxoma virus detected in the Italian hare (Lepus corsicanus)”

 This paper reports the identification and whole genome sequencing of a myxoma virus (MYXV-Ag) infecting an Italian hare (Lepus corsicanus) found dead with clinical signs of myxomatosis (as well as infection by Treponema paraluisleporidarum) in 2018 in Sicily (Italy). This finding is relevant given the current situation in the Iberian Peninsula, after the emergence in 2018 of a new recombinant strain (MYXV-To), causing large severe outbreaks in Iberian hares (Lepus granatensis). The genomic analysis of the Italian strain excluded the presence of MYXV-To and pointed out some genetic differences concerning known MYXV strains.

 I think the paper is clearly written and provides interesting results regarding the issue of myxoma virus infection in hares. I only have minor comments.

 Minor comments:

 - Although it is quite well known in the field of myxoma virus research, I think it would help readers to indicate in the Introduction that myxoma virus Lausanne strain (MYXV-Lu) was deliberately released in France in 1952 from where it spread throughout the whole rabbit range in Europe and became endemic since then. Thus, current strains circulating in Europe derive from this reference strain, after over six decades of field evolution.

 - MYXV-Lu is used as reference strain for the genomic analysis of MYXV-Ag. However, in different parts of the manuscript two different genbank accession numbers (both corresponding to MYXV-Lu) are used: AF170726 and KY548791. The first one corresponds to the original sequencing of Lausanne strain (Cameron et al., Virology 1999, a sequence of 161.773 bp) while the second one is a re-sequence of the same strain (Kerr et al., Plos Pathog 2017, a sequence of 161.777 bp). They are not identical. This should be clarified.

- Lines 220-222: “the known morphological aspects” (“M” and “C”). I think a reference will be useful here.

 - Table 2. There are several genes in this table (i.e., M034L, M096L and M147R) which have a “% aa identity” of 100.0% (between MYXV-Lu and MYXV-Ag) and in the next column show “aa differences”. This seems strange.

 -Lines 292-293: “the M156R, appeared to have significant differences from the strains generally detected in Italy or Europe [28].” Reference 28 deals about Australian virus isolates, not European.

 - Line 294: “In MYXV strains originating from South America”. Maybe it would help to indicate here that MYXV-Lu originally comes from South America (Brazil).

Reviewer 2 Report

Comments and Suggestions for Authors

The authors present a workup of what was likely a myxoma virus infection in an Italian/Corsican hare that was found dead in the hare's endemic area.  The paper is interesting as it is another example of myxoma 'jumping' from rabbits to hares in regions where multiple species of both animals overlap.  The experiments are in-line with what is expected for this type of finding.  The tissue culture, microscopy (EM and IFA) and NGS data are convincing.  The PCR data should be shown as further support that the myxv is unique and not the myxv-To that has been described previously.

Comments on the Quality of English Language

The last recommendation is a quick review of the paper by a native English speaker.  The paper is well written and presented. The language is formal, but the last line of the conclusions, "a new host-adapted stigma" seems out of place.  'Stigma' is commonly used to describe something that is associated with shame that is publicly obvious that sets the thing with the stigma apart and isolated from the norm, so the authors should review their use of 'stigma'.
